# Once Upon a *Time* in *Graph*:
# Relative-Time Pretraining for Complex Temporal Reasoning *

**Sen Yang** [1 2 3]     **Xin Li** [2 3 †]     **Lidong Bing** [2 3]     **Wai Lam** [1]

[1] The Chinese University of Hong Kong

[2] DAMO Academy, Alibaba Group

[3] Hupan Lab, 310023, Hangzhou, China

senyang.stu@gmail.com   {xinting.lx, l.bing}@alibaba-inc.com

wlam@se.cuhk.edu.hk

## Abstract

Our physical world is constantly evolving over time, rendering challenges for pre-trained language models to understand and reason over the temporal contexts of texts. Existing work focuses on strengthening the direct association between a piece of text and its time-stamp. However, the knowledge-time association is usually insufficient for the downstream tasks that require reasoning over temporal dependencies between knowledge. In this work, we make use of the underlying nature of time, all temporally-scoped sentences are strung together through a one-dimensional time axis, and suggest creating a graph structure based on the relative placements of events along the time axis. Inspired by the graph view, we propose REMEMO (Relative Time Modeling), which explicitly connects all temporally-scoped facts by modeling the time relations between any two sentences. Experimental results show that REMEMO outperforms the baseline T5 on multiple temporal question answering datasets under various settings. Further analysis suggests that REMEMO is especially good at modeling long-range complex temporal dependencies. We release our code and pre-trained checkpoints at https://github.com/DAMO-NLP-SG/RemeMo.

## 1 Introduction

Knowledge plays a crucial role in downstream NLP applications. Pre-trained language models (PLMs) generate accurate responses to various user queries by reasoning over their internal knowledge and the knowledge contained in texts.

Real-world knowledge, however, frequently expires and updates. Taking an example in Table 1, the correct answer to the question "*What team did LeBron James play for?*" varies when the temporal

---

**Context:** *LeBron Raymone James Sr. (born December 30, 1984) is an American professional basketball player ······ James was drafted first overall by the* Cleveland Cavaliers in 2003. ······ In 2010, *James famously announced his decision to* join the Miami Heat. ······ *James* returned to the Cleveland Cavaliers in 2014 *and led the team to their first ever NBA championship in 2016, ending the city's 52-year championship drought.* ······

**Conventional Question Answering**

Question (i): *What rank was LeBron James drafted?*

Answer (i): *First overall.*

**Temporal Question Answering**

Question (ii): *What team did LeBron James play for* **in 2003**?

Answer (ii): *Cleveland Cavaliers*.

Question (iii): *What team did LeBron James play for* **in 2012**?

Answer (iii): *Miami Heat*.

Question (iv): *What team did LeBron James play for* **after he left the Miami Heat**?

Answer (iv): *Cleveland Cavaliers*.

Question (v): *What team did LeBron James play for* **when he got into his first NBA Finals**?

Answer (v): *Cleveland Cavaliers*.

Table 1: Examples of conventional QA and temporal QA. In conventional QA, the answer to the question does not change over time. In temporal QA, the answer depends on both the question and the temporal context.

---

context of the question changes. It is thus necessary for PLMs to understand the temporal context of user queries and reason over temporally-scoped facts.

To enhance the temporal understanding capability of PLMs, recent studies have focused on strengthening the one-to-one associations between an event and its corresponding time tokens or time stamps (Dhingra et al., 2022; Rosin and Radinsky, 2022; Rosin et al., 2022). By modeling such superficial temporal dependencies, these methods gain improvements on various time-sensitive tasks.

These methods, however, assume all temporally-scoped facts as independent of each other, ignoring the complex temporal dependencies among

---

* This work was supported by Alibaba Group through the Alibaba Innovative Research (AIR) Program (TA2217728).
† XL is the corresponding author.

temporally-scoped facts. This may pose limitations because time-related queries often require reasoning over multiple temporally-scoped facts. For example, in Table 1, Question (iv) asks "*What team did LeBron James play for after he left the Miami Heat?*" requires reasoning over two facts: (1) James "*join the Miami Heat in 2010*" and (2) James "*returned to the Cleveland Cavaliers in 2014*". By performing the temporal reasoning over path: {"*2014*" is later than "*2010*"; "*returned to the Cleveland Cavaliers*" is associated with "*in 2014*"; "*join the Miami Heat*" is associated with *in 2010*" } ⟹ {"*returned to the Cleveland Cavaliers*" is later than "*join the Miami Heat*"}, Question (iv) can be answered by combining the two facts together. Modeling such complex temporal dependencies benefits downstream time-sensitive tasks, but requires the PLM to understand the concept of time beyond the superficial association between one single fact and its time stamp.

The most well-known concept of time is the linear concept, which views time as a series of moments that unfold sequentially from the past, through the present, and into the future. As shown in Figure 1a, temporally-scoped events are strung together through the one-dimensional time axis. By inspecting the relative positions between any two events, we can build a fully-connected digraph, where events act as graph nodes and temporal relations among them serve as directed graph edges, as shown in Figure 1b. Complex temporal dependencies can thus be systematically modeled from the perspective of graph learning. For example, as shown in Figure 1b, Fact-1 (as the start node) is associated with Fact-2 (as the destination node) either directly through the *earlier* edge or indirectly by passing through two edges via Fact-3.

Inspired by this graph view, we present Relative-Time Modeling (REMEMO): a novel time-aware pre-training framework that exploits relative positions on the time-axis to model complex temporal dependencies. Specifically, we extract time-span tags from sentences and then pre-train the PLM with two joint objectives: (i) the Language Modeling (LM) objective; (ii) a novel Time Relation Classification (TRC) objective, which classifies the relation between any two sentences, i.e., Sentence-A is *earlier*, *later* or *contemporary* [1] compared with Sentence-B. From the graph learning perspec-

tive, the LM objective can be seen as node feature learning, while the TRC objective operates as edge classification.

REMEMO is pre-trained on regular corpora and temporally-scoped sentences extracted from Wikipedia articles. It is then evaluated on seven downstream temporal open-book question answering datasets. REMEMO consistently gives better results compared with the baseline T5 model under multiple settings.

Overall, our contributions are summarized as follows:

- We devise a graph view to represent temporally-scoped events. Grounded on that, we propose TRC, a new pre-training objective that exploits relative time to systematically model complex temporal dependencies.

- We pre-train REMEMO jointly with the LM objective and the proposed TRC objective. Extensive experiments show that REMEMO gains significant improvements over T5 baselines under various settings.

- We present a processing pipeline to obtain normalized time tags for raw texts.

## 2   Related Work

**Temporal Question Answering**   Temporal question answering has attracted much research attention in recent years. An early line of work focused on temporal question answering over knowledge graphs, for which several datasets were built, such as TempQuestions (Jia et al., 2018b), TEQUILA (Jia et al., 2018a), TimeQuestions (Jia et al., 2021) and CRONKGQA (Saxena et al., 2021). Under this temporal KGQA setting, the model is required to do complex temporal reasoning over temporal knowledge graphs and to select an entity node from a given node list as the answer.

More recently, another line of work has focused on temporal question answering either under the closed-book setting to evaluate models' internal memorization of temporal facts or under the open-book setting to evaluate models' temporal understanding and reasoning capability over unstructured texts. Some example datasets include TempLAMA (Dhingra et al., 2022), SituatedQA (Zhang and Choi, 2021), TimeQA (Chen et al., 2021), ArchivalQA (Wang et al., 2022).

---

[1] We omit the *null* label for simplicity, which corresponds to either Sentence-A or Sentence-B does not contain a valid time-span tag.

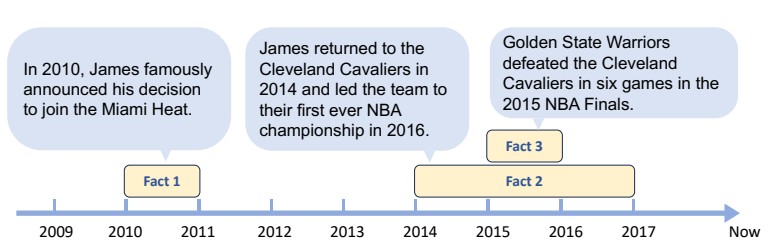

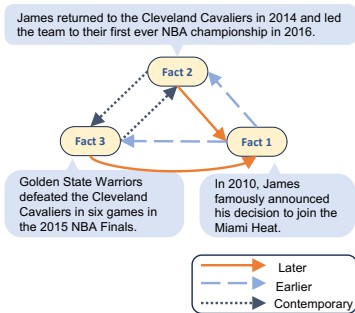

(a) Temporally-scoped facts are strung together through a time-axis.

(b) A fully-connected digraph consisted of facts and time relations.

Figure 1: A fully-connected digraph (Figure 1b) can be constructed based on Figure 1a. In the graph, each piece of fact serves as the head and tail nodes for multiple directed edges - specifically, temporal relations.

StreamingQA (Liska et al., 2022) and TempReason (Tan et al., 2023). Among them, SituatedQA, TimeQA and TempReason are designed to evaluate the temporal understanding and reasoning abilities of the model, so we adopt them to evaluate RE-MEMO.

**Time-aware Modeling** Lots of work has attempted to incorporate time into the modeling process (Wijaya and Yeniterzi, 2011; Hamilton et al., 2016; Bamler and Mandt, 2017; Frermann and Lapata, 2016; Dubossarsky et al., 2019; Giulianelli et al., 2020; Röttger and Pierrehumbert, 2021; Lazaridou et al., 2021; Qin et al., 2021; Luu et al., 2022; Su et al., 2022; Zhao et al., 2022; Cao and Wang, 2022). Among them, some studies that are relevant to this paper explore time-aware pre-training by modeling the one-to-one association between a piece of text and its corresponding time-stamp. For example, Dhingra et al. (2022) pre-pended a document-creation time (e.g., "*year: 2018*") at the beginning of each document so that the PLM better memorizes a fact along with its published time; Rosin et al. (2022) masked out time-informing tokens in a sentence for masked language modeling (MLM) training; Rosin and Radinsky (2022) added a sentence-level time-slot embedding into the self-attention module of Transformer-based models. Different from these methods, our REMEMO is built upon the graph view of time so as to represent complex many-to-many temporal dependencies.

**Event Temporal Relation Extraction** A line of work (Cassidy et al., 2014; Ning et al., 2018; Mathur et al., 2021; Zhang et al., 2022; Huang et al., 2023) has been focusing on event temporal relation extraction (ETRE), which aims to extract the temporal relations between event pairs from text. The main concept shared by our work and previous ETRC works is the temporal relation between two events. However, some task differences exist. Our work focuses on temporal question answering (TQA), in which most clues in existing datasets are explicit and coarse-grained time expressions, such as "*June 2012*", "*18th century*" and "*1980s*". In contrast, the line of ETRE papers focused on time information that is more implicit, fine-grained, and nuanced. Here is an example from Cassidy et al. (2014):

• Example Sentence:
  *Police confirmed Friday that the body found along a highway in San Juan belonged to Jorge Hernandez.*

• TimeBank-Dense Labels:
  *belonged* before *confirmed*; *belonged* before *found*; *found* before *confirmed*; *belonged* before *Friday*; *confirmed* is-included-in *Friday*; *found* is-included-in *Friday*.

## 3 REMEMO

Motivated by the graph view shown in Figure 1b, we present REMEMO to model the temporal dependencies among temporally-scoped sentences[2]. Specifically, given a document containing multiple sentences, we first apply a time-identification & time-normalization pipeline to assign a time-span tag for each sentence (§3.1). Time-relation labels are then created over any pair of temporally-scoped sentences according to the chronological order of the events/facts in the sentences and will be utilized for later time-aware pre-training. Pre-trained on

---

[2]Temporally-scoped sentences are defined as sentences that contain explicit time-informing texts.

both the Language Modeling (LM) objective and the Time Relation Classification (TRC) objective (§3.2), REMEMO is capable of capturing complex temporal dependencies beyond simple time-token co-occurrences. Finally, we discuss variants of RE-MEMO by controlling the graph density of the TRC objective (§3.3).

### 3.1 Pre-processing Pipeline

#### 3.1.1 Time Identification & Normalization

To allow time-aware pre-training, we develop a pipeline to explicitly assign the time tag for each training sentence. The pipeline is comprised of a time token identification model and a rule-based time normalization model. The time-identification model is trained on TimeBank 1.2 (Pustejovsky et al., 2006) and the time-normalization script is adopted from Filannino (2020). Given a document $\mathcal{D}$, we first sentence-tokenize $\mathcal{D}$ into $N$ sentences: $\mathcal{D} = \{\mathcal{S}_1, \mathcal{S}_2, ..., \mathcal{S}_N\}$. Each sentence is fed into the time-identification model to extract time expressions, which works similarly to named-entity recognition, but only for time expressions. Those time expressions are then processed by the time-normalization script to obtain the TimeML (Saurí et al., 2006) type (e.g., date, duration) and value attributes. We further filter out the invalid outputs so that a time tag is assigned for each valid time expression, with a few examples shown in Table 2.

#### 3.1.2 Creating Time-Span Tags

Let *day* be the unit of the time-axis, we propose to map time expression to the corresponding interval of the time-axis and regard such interval as the time-span tag. For instance, the time-span for "June 2012" can be expressed as [2012-06-01, 2012-07-01), where the starting date is inclusive and the ending date is exclusive. For those sentences containing more than one time-expression, we merge all valid time-spans. Note that we merge these time-spans even if they are not strictly consecutive. For example, for the sentence "*In 2003, Tim Duncan won his first NBA Finals MVP award, followed by two more in 2005 and 2007.* ", the time-span tag is [2003-01-01, 2008-01-01), which is the merger of [2003-01-01, 2004-01-01), [2005-01-01, 2006-01-01) and [2007-01-01, 2008-01-01). We adopt this loose strategy because these time-span tags are used to model time relations, for which the leftmost and rightmost time boundaries mostly matter.

One notable defect of this strategy is the loss of granularity. Take the above "*Tim Duncan*" sen-

| Time Expression | Normalized Value |
|---|---|
| *21 July 1924* | 1924-07-21 |
| *November 7, 2006* | 2006-11-07 |
| *June 2012* | 2012-06 |
| *18th century* | 17XX |
| *fourteen fifty* | 1454 |

Table 2: Examples of the extracted time expressions and their normalized values.

tence as an example, the time expression "*2005*" would be ignored if all three time-expressions were merged using our strategy. Generally speaking, if one sentence contains $x$ time-expressions (where $x > 2$), then at least $x - 2$ time-expressions would be ignored by our merging strategy. As for our pre-training corpora, roughly 75% of all temporally-scoped sentences have $\leq 2$ time-expressions per sentence, so the other 25% would see losses of granularity.

We include more details and manually verify the reliability of our pre-processing pipeline in Appendix § A.1.

### 3.2 Pre-training Framework

As shown in Figure 1a, temporally-scoped sentences are strung together through the one-dimensional time-axis, on which each sentence covers a specific time-span. A fully-connected digraph can thus be constructed to connect all these sentences by inspecting the relative positions of any two time-spans, as shown in Figure 1b. Inspired by this graph view, we propose the Time Relation Classification (TRC) objective to simulate the graph structure with textual inputs.

**Creating TRC Input Instances** To simulate a graph consisting of multiple nodes, we put multiple temporally-scoped sentences into one context, where sentences act as nodes and the whole context acts as the graph. Because nodes in the graph are clearly separated, we mark the boundaries of these sentences with special tokens so that the model can correctly identify each sentence (node). Specifically, we insert two temporal special tokens, [TIME] and [/TIME], at the beginning and at the end of each sentence, respectively. Overall, given $M$ temporally-scoped sentences $\{\mathcal{S}_1^{\mathrm{T}}, ..., \mathcal{S}_M^{\mathrm{T}}\}$, one training instance looks like:  [TIME] $\mathcal{S}_1^{\mathrm{T}}$ [/TIME] [TIME] $\mathcal{S}_2^{\mathrm{T}}$ [/TIME] ... [TIME] $\mathcal{S}_M^{\mathrm{T}}$ [/TIME] .

**TRC Objective** The aforementioned fully-connected digraph is constructed by inspecting the relative positions of any two temporally-scoped sentences in the same context. These relative positions act as edges in the graph shown in Figure 1b. The directed edge from Sentence-$i$ to Sentence-$j$ may lie in one of the three categories: (i) *Earlier*: The right time-span boundary of Sentence-$i$ is smaller than the left time-span boundary of Sentence-$j$; (ii) *Later*: The left time-span boundary of Sentence-$i$ is larger than the right time-span boundary of Sentence-$j$; (iii) *Contemporary*: The time-spans of Sentence-$i$ and Sentence-$j$ overlap. These directed edges are adopted as training labels for the TRC objective. Formally, given a training instance that simulates the graph structure, `` [TIME] $\mathcal{S}_1^{\mathrm{T}}$ [/TIME] ... [TIME]$\mathcal{S}_i^{\mathrm{T}}$ [/TIME] ... [TIME] $\mathcal{S}_j^{\mathrm{T}}$ [/TIME] ... [TIME] $\mathcal{S}_M^{\mathrm{T}}$ [/TIME] ``, the TRC objective aims to classify the directed edge $r_{ij}$ connecting $\mathcal{S}_i^{\mathrm{T}}$ to $\mathcal{S}_j^{\mathrm{T}}$, where $r_{ij} \in \{$*Earlier*, *Later*, *Contemporary*$\}$, $\forall i, j \in \{1, ..., M\}$ and $i \neq j$. Taking the adjacency matrix shown in Figure 2 as an example, $r_{12} = $ *Earlier* and $r_{32} = $ *Contemporary*. In practice, we adopt the representation of the [TIME] special token positioning at the start of $\mathcal{S}_i^{\mathrm{T}}$ as the temporal representation for $\mathcal{S}_i^{\mathrm{T}}$, and denote it by $\mathbf{h}_i^{\mathtt{[TIME]}}$. We concatenate the two representations, $\mathbf{h}_i^{\mathtt{[TIME]}}$ and $\mathbf{h}_j^{\mathtt{[TIME]}}$, to represent the directed edge pointing from Node $\mathcal{S}_i^{\mathrm{T}}$ to Node $\mathcal{S}_j^{\mathrm{T}}$. The concatenated representation is then fed into a two-layer MLP to predict $r_{ij}$. We build REMEMO based on T5 checkpoints (Raffel et al., 2020) and apply the TRC objective to T5 encoder hidden states. The TRC loss function is written as

$$\mathcal{L}_{\mathrm{TRC}} = -\sum_{i=1}^{M} \sum_{\substack{j=1 \\ j \neq i}}^{M} \log p \left( r_{ij} \middle| \left[ \mathbf{h}_i^{\mathtt{[TIME]}}; \mathbf{h}_j^{\mathtt{[TIME]}} \right] \right) \tag{1}$$

where $M$ is the total number of temporally-scoped sentences in the context. $[\circ; \circ]$ is the vector concatenation operator.

**Pre-training** Two objectives are adopted for pre-training REMEMO, including the Language Modeling (LM) objective and the Time Relation Classification (TRC) objective. Since we adopt T5 architecture, the LM objective is the same as that of T5, i.e., the denoising objective. The LM objective applies to all instances, while the TRC objective applies to temporal instances that we formulate earlier. Overall, the loss function is:

$$\mathcal{L} = \mathcal{L}_{\mathrm{LM}} + \mathcal{L}_{\mathrm{TRC}} \tag{2}$$

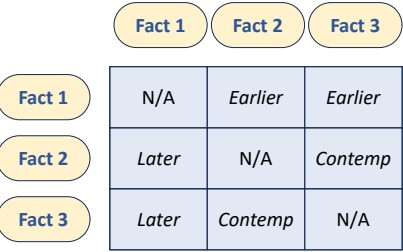

|        | Fact 1 | Fact 2 | Fact 3 |
|--------|--------|--------|--------|
| Fact 1 | N/A    | Earlier | Earlier |
| Fact 2 | Later  | N/A    | Contemp |
| Fact 3 | Later  | Contemp | N/A    |

Figure 2: TRC labels (i.e., adjacency matrix) for the example graph in Figure 1b. *Contemp* refers to *Contemporary*.

where

$$\mathcal{L}_{\mathrm{LM}} = -\sum_i \log p(x_i | \mathbf{h_i}) \tag{3}$$

where $x_i$ is the $i$-th token of the input instance and $\mathbf{h}_i$ is the corresponding token representation.

### 3.3 Graph Density

The density of a graph is defined to be the ratio of the number of edges $|E|$ with respect to the maximum possible edges. In the case of digraph, the density is

$$D = \frac{|E|}{|V|(|V| - 1)} \tag{4}$$

where $|V|$ is the number of vertices (nodes).

As shown in Figures 1b and 2, given $n$ graph nodes (i.e., temporally-scoped sentences) in the digraph, there would be $n(n - 1) \simeq \mathcal{O}(n^2)$ directed edges (e.g., time relations of Sentence-$i$ to Sentence-$j$). In other words, the vanilla TRC objective adopts a graph density $D = 1.0$. It thus raises an interesting question of how graph density influences model performance. In § 4.5, we empirically investigate variants of REMEMO by controlling the graph density to be $D = \frac{\log n}{n}$ and $D = \frac{1}{n}$, respectively.

## 4 Experiments

We present the empirical study in this section. Experimental settings are in § 4.1 and § 4.2. Main results are in § 4.3. We further investigate the challenges of modeling complex temporal dependencies in § 4.4.

### 4.1 Pre-training Setup

#### 4.1.1 Pre-processing

We adopt the pre-trained RoBERTa-base (Liu et al., 2019) checkpoint to initialize our time-identification model and further fine-tune it on TimeBank 1.2 (Pustejovsky et al., 2006). As for

| | TimeQA | | | | TempReason | | | | | | | | SituatedQA | | Avg. | |
| | Easy | | Hard | | ReasonQA L2 | | OBQA L2 | | ReasonQA L3 | | OBQA L3 | | | | | |
| | F1 | EM | F1 | EM | F1 | EM | F1 | EM | F1 | EM | F1 | EM | F1 | EM | F1 | EM |
|---|---|---|---|---|---|---|---|---|---|---|---|---|---|---|---|---|
| T5$_{base}$ | 68.23 | 59.99 | 64.06 | 55.56 | 90.62 | 87.56 | 44.99 | 25.96 | 59.56 | 48.92 | 41.76 | 23.79 | 52.55 | 45.72 | 60.25 | 49.64 |
| T5$_{base}$ + LM | 69.81 | 61.06 | 65.71 | 56.99 | 94.64 | 92.34 | 49.03 | 31.02 | 62.71 | 53.91 | 43.02 | 26.11 | 54.07 | 47.19 | 62.71 | 52.66 |
| + TempT5 | – | – | – | – | 95.05 | 92.67 | 49.60 | 31.77 | 63.92 | 54.73 | 43.00 | 26.06 | – | – | – | – |
| REMEMO$_{base}$ | **70.35** | **61.36** | **67.28** | **58.19** | **97.67** | **96.22** | **51.58** | **33.62** | **91.17** | **89.29** | **44.91** | **28.48** | **55.06** | **48.12** | **68.29** | **59.33** |
| T5$_{large}$ | 71.60 | 63.10 | 68.11 | 59.49 | 96.68 | 94.80 | 50.91 | 32.72 | 60.24 | 50.21 | 46.78 | 28.84 | 55.56 | 48.77 | 64.27 | 53.99 |
| T5$_{large}$ + LM | 71.72 | 62.76 | **70.59** | **61.89** | 97.78 | 96.17 | 53.73 | 35.98 | 63.50 | 55.41 | 48.26 | 31.93 | 54.71 | 47.76 | 65.76 | 55.99 |
| REMEMO$_{large}$ | **72.25** | **63.73** | 69.31 | 60.49 | **98.15** | **96.74** | **54.94** | **37.40** | **94.02** | **92.69** | **49.31** | **33.37** | **56.06** | **49.12** | **70.58** | **61.93** |
| GPT-3.5-turbo* | 53.14 | 44.00 | 26.32 | 19.00 | 32.83 | 28.00 | 9.52 | 6.00 | 44.30 | 36.00 | 22.01 | 7.00 | 17.19 | 7.00 | 29.33 | 21.00 |
| GPT-4* | 56.19 | 48.00 | 35.90 | 29.00 | 90.15 | 84.00 | 36.84 | 19.00 | 82.15 | 76.00 | 46.35 | 29.00 | 60.79 | 54.00 | 58.34 | 48.43 |

Table 3: Main results under the single-context setting. The datasets include TimeQA (two splits: {Easy, Hard}), TempReason (four splits: {Reason Level-2, OBQA Level-2, Reason Level-3, OBQA Level-3}) and SituatedQA. We report both F1 and exact match (EM) scores. LLM results (with *) are evaluated on 100 randomly-sampled test-set instances and are thus not directly comparable to the full-size test-set results.

time normalization, we only keep those predicted values that are valid time spans. We manually verify the high accuracy of our pre-processing pipeline, with details shown in Appendix § A.1.

### 4.1.2 Pre-training Implementation

We adopt the pre-trained T5 v1.1 (Raffel et al., 2020) checkpoints to initialize our model and perform continual pre-training on Wikipedia and Bookcorpus. The peak learning rates are set to 5e-5 and 3e-4 for base-sized and large-sized models, respectively. We follow the pre-training of T5 to take Adafactor (Shazeer and Stern, 2018) as the optimizer with a batch size of 2048 and parameter-scaling enabled across all the experiments. We pre-train all models for 8,000 steps, take a warm-up ratio of 10% to linearly increase the learning rate, and adopt the cosine learning rate scheduler after warm-up. We implement our code with Hugging-Face Transformers library (Wolf et al., 2020). It takes about 60 hours to train the base-size model on 4 Nvidia V100-32GB GPUs and about 75 hours to train the large-size model on 2 Nvidia A100-80GB GPUs with BF16.

### 4.2 Evaluation Setup

We evaluate REMEMO on seven downstream temporal QA datasets and under three fine-tuning settings. We report both F1 and exact match (EM) for all experiments, using the evaluation code of SQuAD (Rajpurkar et al., 2016) and SQuAD-v2 (Rajpurkar et al., 2018).

### 4.2.1 Datasets

We give a brief description of our adopted datasets in this section.

| | TimeQA | | | | Avg. | |
| | Easy | | Hard | | | |
| | F1 | EM | F1 | EM | F1 | EM |
|---|---|---|---|---|---|---|
| T5$_{base}$ | 63.65 | 54.19 | 50.21 | 41.16 | 56.93 | 47.675 |
| T5$_{base}$ + LM | 65.61 | 55.92 | 57.46 | 48.05 | 61.54 | 51.99 |
| REMEMO$_{base}$ | **66.14** | **56.56** | **59.84** | **50.71** | **62.99** | **53.64** |
| (Hyper-parameter Search on TimeQA-Hard) | | | | | | |
| T5$_{large}$ | 52.15 | 42.48 | 62.15 | 53.09 | 57.15 | 47.79 |
| T5$_{large}$ + LM | 68.90 | 59.46 | 62.54 | 53.48 | 65.72 | 56.47 |
| REMEMO$_{large}$ | **69.53** | **60.26** | **63.67** | **54.84** | **66.60** | **57.55** |

Table 4: Fusion-in-Decoder (FiD) results on TimeQA (two splits: {Easy, Hard}). Note that the large-scale hyper-parameter search is performed on TimeQA-Hard and then directly applied to TimeQA-Easy, due to the high computational cost of FiD.

**TimeQA** TimeQA (Chen et al., 2021) is designed to evaluate the model's reasoning capability of time-evolving facts. This dataset contains two difficulty levels: The easy-level split tends to be answered based on surface form rather than temporal reasoning while the hard-level split is more likely to necessitate reasoning over the implicit time information.

**SituatedQA** SituatedQA (Zhang and Choi, 2021) contains questions where extra-linguistic contexts need to be considered, e.g., depending on the temporal or geographical context. We adopt their temporal-dependent dataset for our experiments.

**TempReason** TempReason (Tan et al., 2023) is designed to evaluate models' temporal reasoning ability. It consists of two levels (i.e., L-2: "time-event" time relation; L-3:"event-event" time-relation) and two settings (i.e., OBQA: conven-

Table 5 part 1:

| | | TimeQA | | | | TempReason | | | |
| | | Easy | | Hard | | ReasonQA L2 | | OBQA L2 | |
| | | F1 | EM | F1 | EM | F1 | EM | F1 | EM |
|---|---|---|---|---|---|---|---|---|---|
| **16-Shot** | T5$_{base}$ | $8.02_{\pm1.84}$ | $1.95_{\pm2.25}$ | $8.95_{\pm1.60}$ | $2.78_{\pm3.22}$ | $14.60_{\pm1.50}$ | $0.00_{\pm0.00}$ | $8.17_{\pm3.05}$ | $0.09_{\pm0.08}$ |
| | T5$_{base}$ + LM | $\mathbf{14.30}_{\pm3.86}$ | $9.88_{\pm2.64}$ | $\mathbf{15.51}_{\pm2.19}$ | $\mathbf{12.64}_{\pm1.17}$ | $14.39_{\pm3.62}$ | $0.31_{\pm0.39}$ | $\mathbf{16.69}_{\pm2.97}$ | $\mathbf{3.07}_{\pm0.59}$ |
| | REMEMO$_{base}$ | $14.05_{\pm1.99}$ | $\mathbf{11.21}_{\pm1.83}$ | $14.77_{\pm2.13}$ | $13.85_{\pm2.48}$ | $\mathbf{23.12}_{\pm10.41}$ | $\mathbf{9.96}_{\pm8.31}$ | $14.06_{\pm4.57}$ | $2.98_{\pm1.25}$ |
| **64-Shot** | T5$_{base}$ | $12.37_{\pm2.66}$ | $2.92_{\pm2.19}$ | $11.79_{\pm0.87}$ | $3.31_{\pm3.02}$ | $24.87_{\pm2.01}$ | $2.02_{\pm1.22}$ | $21.20_{\pm4.76}$ | $2.83_{\pm1.83}$ |
| | T5$_{base}$ + LM | $21.61_{\pm2.45}$ | $14.02_{\pm2.14}$ | $\mathbf{24.11}_{\pm3.00}$ | $\mathbf{16.04}_{\pm1.96}$ | $39.21_{\pm2.68}$ | $24.99_{\pm3.46}$ | $23.72_{\pm0.57}$ | $5.44_{\pm0.85}$ |
| | REMEMO$_{base}$ | $\mathbf{26.54}_{\pm6.72}$ | $\mathbf{18.20}_{\pm5.33}$ | $22.11_{\pm3.50}$ | $14.57_{\pm2.61}$ | $\mathbf{55.47}_{\pm11.87}$ | $\mathbf{42.17}_{\pm13.99}$ | $\mathbf{26.43}_{\pm2.75}$ | $\mathbf{7.12}_{\pm2.59}$ |

Table 5 part 2:

| | | TempReason | | | | SituatedQA | | Avg. | |
| | *(Continued)* | ReasonQA L3 | | OBQA L3 | | | | | |
| | | F1 | EM | F1 | EM | F1 | EM | F1 | EM |
|---|---|---|---|---|---|---|---|---|---|
| **16-Shot** | T5$_{base}$ | $15.48_{\pm5.16}$ | $0.57_{\pm1.13}$ | $\mathbf{13.02}_{\pm4.54}$ | $0.15_{\pm0.31}$ | $3.16_{\pm1.02}$ | $0.16_{\pm0.21}$ | $10.20_{\pm2.67}$ | $0.82_{\pm1.03}$ |
| | T5$_{base}$ + LM | $23.89_{\pm3.07}$ | $9.42_{\pm3.98}$ | $12.10_{\pm5.21}$ | $\mathbf{1.19}_{\pm1.00}$ | $3.30_{\pm0.54}$ | $0.09_{\pm0.07}$ | $14.31_{\pm3.06}$ | $5.23_{\pm1.41}$ |
| | REMEMO$_{base}$ | $\mathbf{24.69}_{\pm8.19}$ | $\mathbf{10.83}_{\pm6.77}$ | $11.18_{\pm4.62}$ | $1.11_{\pm1.13}$ | $\mathbf{3.73}_{\pm0.80}$ | $\mathbf{0.28}_{\pm0.13}$ | $\mathbf{15.09}_{\pm4.67}$ | $\mathbf{7.17}_{\pm3.13}$ |
| **64-Shot** | T5$_{base}$ | $18.60_{\pm5.96}$ | $0.80_{\pm0.58}$ | $20.80_{\pm4.50}$ | $3.02_{\pm1.65}$ | $6.12_{\pm3.38}$ | $0.37_{\pm0.18}$ | $16.54_{\pm3.45}$ | $2.18_{\pm1.53}$ |
| | T5$_{base}$ + LM | $43.10_{\pm5.71}$ | $29.40_{\pm5.15}$ | $24.51_{\pm1.63}$ | $4.37_{\pm0.99}$ | $\mathbf{26.64}_{\pm6.91}$ | $\mathbf{20.64}_{\pm6.18}$ | $28.99_{\pm3.28}$ | $16.41_{\pm2.96}$ |
| | REMEMO$_{base}$ | $\mathbf{44.79}_{\pm5.61}$ | $\mathbf{31.38}_{\pm6.58}$ | $\mathbf{25.50}_{\pm2.45}$ | $\mathbf{5.99}_{\pm1.87}$ | $26.46_{\pm7.35}$ | $20.38_{\pm6.77}$ | $\mathbf{32.47}_{\pm5.75}$ | $\mathbf{19.97}_{\pm5.68}$ |

Table 5: Few-shot results under the setting of single-context fine-tuning. We report average F1 and exact match (EM) scores along with their standard deviations across five runs with five different random seeds.

tional open-book QA; ReasonQA: the simplified version of OBQA by concatenating all relevant facts into a paragraph), rendering four splits in total. Note that our reported baseline results are different from those reported in Tan et al. (2023) due to a few differences in implementations. More details are shown in Appendix § A.5.

Example instances of the above datasets can be found in Appendix § A.3

### 4.2.2 Fine-tuning Settings

We adopt three settings to evaluate REMEMO: (1) single-context fine-tuning, (2) few-shot fine-tuning, and (3) Fusion-in-Decoder (FiD) fine-tuning. All 7 datasets are evaluated under the single-context and the few-shot settings, while two of these datasets are included in the FiD setting. All models are optimized with AdamW (Loshchilov and Hutter, 2019). We perform hyper-parameter search in all experiments except FiD of large-sized models. More details are shown in Appendix § A.2.

**Single-Context** Under the single-context setting, the context and the question are concatenated together as the input to the model and the model is required to generate the answer by comprehending the context. The maximum input length is set to 512 after tokenization. To keep the input length lower than 512, we truncate the context to 1500 chars while ensuring the correct answer occurs at least once in the truncated context.

**Fusion-in-Decoder** Since Fusion-in-Decoder (FiD) (Izacard and Grave, 2021) was adopted as a standard baseline in the original paper of TimeQA, we also conduct FiD experiments for a comprehensive comparison.

**Few-shot** We conduct few-shot experiments to investigate the ability of REMEMO to efficiently adapt to time-related QA tasks. For a $k$-shot setting, we randomly sample $k$ training and development instances, respectively, while the test set remains the same. We run experiments for 16-shot and 64-shot.

### 4.2.3 Baselines

**T5** stands for the vanilla T5 checkpoints, which are directly applied for supervised fine-tuning.

**T5 + LM** is pre-trained using the LM objective. The only difference between REMEMO and T5 + LM is that REMEMO adopts the TRC objective but T5 + LM does not.

**TempT5** (Tan et al., 2023) is a reinforcement learning (RL) method that further improves supervised fine-tuned (SFT) T5 models. It requires the dataset to have negative answer candidates, so it can only be applied to TempReason (Tan et al., 2023). It is added upon a fine-tuned checkpoint, so

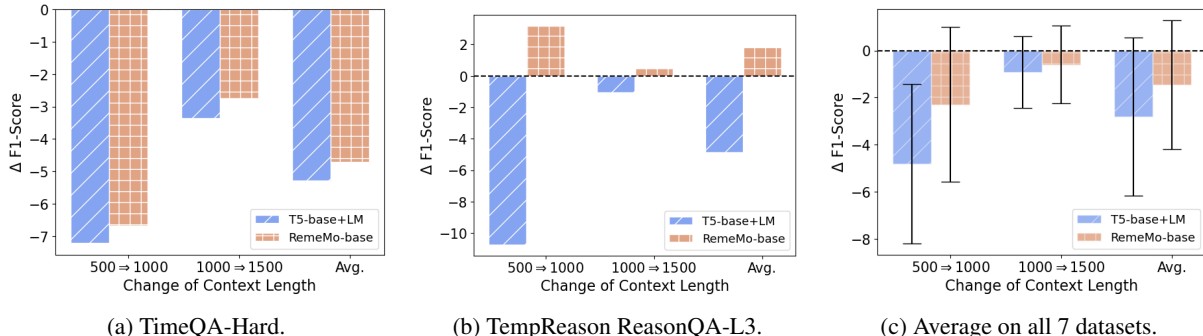

| (a) TimeQA-Hard. | (b) TempReason ReasonQA-L3. | (c) Average on all 7 datasets. |

Figure 3: Changes in F1-scores in response to the changes in context lengths (i.e., $500 \Rightarrow 1000$, $1000 \Rightarrow 1500$, and the average of the two). Blue bars with slash "/" are $\text{T5}_{\text{base}}$+LM, while orange bars with cross "+" are $\text{REMEMO}_{\text{base}}$. Positive $\Delta$F1-scores indicate that the model obtains improvements as contexts become longer, while negative $\Delta$F1-scores indicate declines with longer contexts. Standard deviations on all 7 datasets are also reported for Figure 3c.

| | TimeQA | | | | TempReason | | | | | | | | SituatedQA | | Avg. | |
| | Easy | | Hard | | ReasonQA L2 | | OBQA L2 | | ReasonQA L3 | | OBQA L3 | | | | | |
| | F1 | EM | F1 | EM | F1 | EM | F1 | EM | F1 | EM | F1 | EM | F1 | EM | F1 | EM |
|---|---|---|---|---|---|---|---|---|---|---|---|---|---|---|---|---|
| $D = 1.0$ | **70.35** | **61.36** | 67.28 | 58.19 | **97.67** | **96.22** | 51.58 | 33.62 | **91.17** | **89.29** | 44.91 | **28.48** | 55.06 | 48.12 | **68.29** | **59.33** |
| $D = (\log n)/n$ | 69.86 | 61.03 | 67.25 | 58.32 | 97.63 | 95.84 | **51.69** | **33.85** | 90.09 | 88.16 | 44.76 | 26.83 | **55.36** | **48.19** | 68.09 | 58.89 |
| $D = 1/n$ | 69.48 | 60.56 | **67.92** | **58.93** | 97.03 | 95.37 | 50.71 | 33.33 | 89.54 | 87.33 | 44.49 | 28.32 | 54.49 | 47.55 | 67.67 | 58.77 |
| $\text{T5}_{\text{base}}$ + LM | 69.81 | 61.06 | 65.71 | 56.99 | 94.64 | 92.34 | 49.03 | 31.02 | 62.71 | 53.91 | 43.02 | 26.11 | 54.07 | 47.19 | 62.71 | 52.66 |
| $\text{T5}_{\text{base}}$ | 68.23 | 59.99 | 64.06 | 55.56 | 90.62 | 87.56 | 44.99 | 25.96 | 59.56 | 48.92 | 41.76 | 23.79 | 52.55 | 45.72 | 60.25 | 49.64 |

Table 6: Ablation results on graph density of the TRC objective. We modify the graph density to be $D = 1.0$, $D = \frac{\log n}{n}$ and $D = \frac{1}{n}$, respectively, where $n$ is the number of temporally-scoped sentences in a training instance.

we can directly apply it on all other baselines (T5 and T5 + LM) and REMEMO.

**GPT-3.5-turbo & GPT-4** are two powerful large language models (LLMs). We evaluate their zero-shot performances on 100 randomly-sampled testing instances under the single-context setting due to budget limit. The prompt templates and the model versions that we adopt are shown in Appendix § A.4.

### 4.3 Main Results

**Single-Context** Table 3 shows the single-context results on all seven datasets. REMEMO outperforms T5 on most datasets, with an average improvement of +5.58 F1-scores over $\text{T5}_{\text{base}}$ + LM and +4.82 over $\text{T5}_{\text{large}}$ + LM. TempT5 improves the performance of T5 + LM on most splits but does not surpass REMEMO. Notably, REMEMO outperforms T5 for large margins with both base and large scales on ReasonQA Level-3, on which the model is required to reason over "event-event" temporal relations. This result suggests that the TRC training objective is effective for modeling complex temporal dependencies.

**Fusion-in-Decoder** Table 4 shows the FiD results on TimeQA-Easy & -Hard. REMEMO outperforms the baseline T5 on both datasets. The average improvements in terms of EM are +1.65 for the base scale and +1.08 for the large scale.

**Few-shot** Table 5 shows the 16- and 64-shot results under the single-context setting with the base-scale models. $\text{REMEMO}_{\text{base}}$ outperforms the strongest baseline on most datasets, with average improvements of +1.9 EM under 16-shot and +3.6 EM under 64-shot. Notably, $\text{REMEMO}_{\text{base}}$ achieves performance gains more rapidly compared with $\text{T5}_{\text{base}}$+LM ($\frac{32.47}{15.09} \approx 215.2\%$ versus $\frac{28.99}{14.31} \approx 202.6\%$ in terms of relative F1), when the number of available instances increases from 16 to 64. These results suggest that REMEMO is more effective to reach peak performance with fewer instances.

### 4.4 Better Modeling of Complex Temporal Dependencies

One crucial challenge faced by open-book QA systems is how to connect the question with relevant sentences in the context while not being misguided

by irrelevant ones. This problem becomes more severe when the context becomes longer because the number of possible dependencies among context sentences could increase enormously, rendering more complexity when dealing with those dependencies.

As for temporal question answering, the most practical dependencies are mostly temporal dependencies. To investigate how our model handles much more complex temporal dependencies, we modify the context lengths to be 1,000 and 500 chars, respectively, in addition to the regular setup of 1,500 chars under the single-context setting. Figures 3a, 3b and 3c show the changes in F1-scores of $T5_{base}$+LM and $REMEMO_{base}$ on TimeQA-Hard, on TempReason ReasonQA-L2 and on the average of all 7 datasets when context lengths increase. On average, the F1-score decrease of $REMEMO_{base}$ is much lower than that of $T5_{base}$+LM. On TempReason ReasonQA-L3, longer contexts hurt the performance of $T5_{base}$+LM, but surprisingly boost the performance of $REMEMO_{base}$. The relatively smaller F1-score performance drop (or even improvement) of REMEMO suggests that REMEMO can better models long-range and complex temporal dependencies.

### 4.5   Ablation Study: Graph Density

Table 6 shows the ablation results on graph density of the TRC objective (§ 3.3). In addition to the regular setting (i.e., $D = 1.0$), we pre-train another two $REMEMMO_{base}$ variants with different graph densities by controlling the number of edges in the digraph (i.e., $D = \frac{\log n}{n}$ and $D = \frac{1}{n}$). As the TRC training labels become more sparse ($1.0 \rightarrow \frac{\log n}{n} \rightarrow \frac{1}{n}$), downstream performances on temporal QA show mild decreases on most datasets. All three variants, however, still outperform $T5_{base}$ baselines significantly. These results suggest that the key to the superior performances of REMEMO is the graph-like modeling of temporally-scoped events rather than the density of graph edges.

## 5   Conclusions

In this paper, we devised a graph view of temporally-scoped events based on the fact that these events are strung together through the one-dimensional time axis. Inspired by this graph view, we proposed the TRC pre-training objective that exploits relative time as directed graph edges for complex temporal reasoning. We pre-trained RE-

MEMO jointly with the LM objective and the TRC objective and evaluated it on seven downstream temporal QA datasets. Experimental results show that REMEMO outperforms the baseline T5 under various settings. The performance improvements are remarkably large on datasets that require reasoning over complex "event-event" temporal relations. Further analysis suggests that (i) REMEMO is especially good at modeling long-range complex temporal dependencies; (ii) The key to the success of REMEMO is the graph-inspired modeling rather than the larger density of the graph.

## Limitations

We observe two limitations regarding our work:

- Because the TRC objective requires time-span tags of textual sentences, we implement a pre-processing pipeline (§3.1), which contains a rule-based method to normalize time expressions. This may limit our method from generalizing to other languages since each language requires its corresponding time-normalizaton method.

- The TRC objective is designed to simulate the graph-like structure, as described in §3.2. However, the order of the graph (i.e., the number of nodes in the graph) is limited by the length of the context, restricting us from increasing the order of the graph to arbitrarily large.

## Ethics Statement

In this paper, we adopt Wikipedia and BookCorpus for pre-training the language model. These two corpora are publicly available and widely used for research purposes in the NLP area. We adopt TimeQA, SituatedQA and TempReason for evaluation. These datasets are all publicly available and are for research purposes only. However, these corpora/datasets may still contain improper or harmful content. None of such content reflects the opinions of the authors of this paper.

## Acknowledgements

We thank Qingyu Tan for sharing TempReason Dataset. We thank the anonymous reviewers for their thorough and constructive suggestions that help make this work better.

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

# A  Appendix

## A.1  Details and Reliability of Pre-processing Pipeline

As for the time-identification model, we train it with a learning-rate to be 2e-5 and a batch-size to be 32.

We generally adopt aggressive filtering during pre-processing. Specifically, we focus on two parts:

- Filtering out invalid time-identification tags produced by the fine-tuned RoBERTa model. Specifically, following the NER-like tagging manner, our time-identification model predicts a tag among {B-TIME, I-TIME, O} for each token. We exclude all sequences of tokens that start with I-TIME (a valid NE should start with B-TIME), even though these sequences starting with I-TIME still might be valid time expressions in some cases.

- Filtering out time-normalization results that are somehow unlikely to be valid dates. Below we show a few examples of what we included and excluded. Each example is a tuple containing four elements: (temporal expression, TYPE {DATE, TIME, SET, DURATION} according to TimeML, normalized VALUE according to TimeML, specific format). We only focus on tuples with TYPE=DATE. Our filtering strategy works by checking the specific format.

  - Included:
    * ('2015', 'DATE', '2015', 'Year')
    * ("the early ' 90s", 'DATE', '199X', '90s nineties')
    * ('may 1913', 'DATE', '1913-05', 'Month Year')
    * ('november 1 , 2015', 'DATE', '2015-11-01', 'Month. Date, MM DD YYYY')
    * ('30 november 1945', 'DATE', '1945-11-30', 'Day Month Year mic')
    * ('19th century', 'DATE', '18XX', '20th century')
    * ('01/01/2022', 'DATE', '2022-01-01', 'DCT3')
    * ('seventeen hundred and fifty two', 'DATE', '1752', 'Year number 90s&before')

∗ ('2043/11/05', 'DATE', '2043-11-05', 'mic1')
- Excluded:
  ∗ ('031300010703', 'DATE', '0313-00-01', 'DCT1mic')
    · Unlikely to be time expressions.
  ∗ ('2019/11/25', 'DATE', '25-11-19', 'DCT4')
    · Ambiguous: may be 2019/11/25 or 2025/11/19.
  ∗ ('2004 third quarter', 'DATE', '2004-Q3', 'year quarter')
    · By manually checking, we found that the results of year quarter format often contain errors.

The above strategies are mainly supported by manual qualitative evaluations conducted by the authors of this paper.

| Averaged Precision | Averaged Recall | Averaged F1 | Averaged EM |
|---|---|---|---|
| 0.93 | 0.89 | 0.84 | 0.76 |

Table 7: Reliability evaluation of our pre-processing pipeline on 100 randomly-sampled instances.

Besides, to measure the accuracy of our pre-processing pipeline, we randomly sample 100 sentences that are predicted to contain valid time-span tags, manually annotate the time-span tags, and compare those tags with the automatically predicted ones. We adopt the overlap between the annotated and the predicted time-spans in terms of *day* to calculate "F1-score" [3]. As can be seen from Table 7, the average precision is 0.93, which indicates the effectiveness of our strategy.

## A.2 Evaluation Settings

**Single-Context** The maximum input length is set to 512 after tokenization. We adopt hyper-parameter search to find out the best setting for each dataset: batch-size = {16, 32}; learning-rate = {3e-5, 1e-4, 3e-4}. We train all models for 10 epochs, use a warm-up ratio of 10% for the -large-size experiments and no warm-up for the -base size, and adopt the linear learning rate scheduler. During fine-tuning, the model is evaluated on the development set every half epoch. We report test

---

[3] For example, suppose the annotated time-span is [2023-01-01, 2023-01-24] and the predicted time-span is [2023-01-01, 2023-02-01], then precision $= \frac{23}{31} = 74.2\%$, recall $= 100\%$ and F1-score $= 85.2\%$.

set results of the models obtaining the best development set F1-scores. For TimeQA, we adopt the value of key "*context*" in the released JSON files as context. For SituatedQA, since their officially released dataset files do not contain contexts, we follow the authors to adopt DPR (Karpukhin et al., 2020) to retrieve contexts for each question from the Wikipedia Dump of 2021-Feb-20. We concatenate the top-2 retrieved snippets to formulate one context.

**Fusion-in-Decoder** For the base-size models, we perform hyper-parameter search: batch-size = {32, 128}; learning-rate = {5e-5, 1e-4, 1e-3}. For the large-size models, due to the heavy computational cost of FiD, we only perform hyper-parameter search for TimeQA-Hard to obtain the best setting (lr=1e-4, bsz=32) and directly apply it to the other two datasets. We train all models for 5 epochs, linearly warm up the model for the initial 10% steps, and then linearly decrease the learning rate. The maximum number of contexts is set to 100 and each context contains 250 tokens at most. We report the test set results of the models that obtain the best development set results. For TimeQA, we adopt the value of the key "*paragraphs*" in the released JSON files as FiD contexts.

**Few-shot** We perform hyper-parameter search (batch-size = {8, 64}; learning-rate = {1e-4, 5e-4}) and run each setting five times with five different random seeds to minimize the effects of randomness. We use AdamW to train all models for 10 epochs with the linear learning rate scheduler.

## A.3 Example Data Points

Example data instances are shown in Tables 8, 9 and 10. We underline those context sentence(s) that contain the necessary information to answer the question.

## A.4 LLM Settings

Our LLM experiments were conducted in June 2023. We adopt GPT-4-32K. We filter out responses like "*The context does not provide information on which employer Corine Mauch worked for in Jun 2008.*" and replace them with empty strings by manually writing a few simple rules.

We write our prompts based on the prompts of Li et al. (2023). For datasets that do not contain null answers (i.e., all four splits of TempReason and SituatedQA), we use the following prompt:

```
PROMPT_TEMPLATE = \
    """

  Answer the question based on the context.
   Only answer the name.

   If there are more than one answer,
  only give me the most suitable answer.

   \nContext: {}
   \nQuestion: {}
   \nAnswer:
    """
```

For TimeQA-Hard and TimeQA-Easy, which include null answers, we use the following prompt:

```
PROMPT_TEMPLATE_V2 = \
    """

  Answer the question based on the context.
   Only answer the name.

   If there are more than one answer,
  only give me the most suitable answer.

   If the question cannot be answered
  based on the context, answer \"No Answer\".

   \nContext: {}
   \nQuestion: {}
   \nAnswer:
    """
```

### A.5   Differences in Baseline Implementations

The different baseline performances between this work and Tan et al. (2023) result from a few differences in implementations.

**T5 Versions**   As mentioned in § 4.1, we adopted T5-v1.1 for all experiments, while the original TempReason papers adopted T5-v1.0. We did not adopt T5-v1.0 because its pre-training corpora included downstream datasets, such as SQuAD, which makes the evaluation of few-shot QA unfair. In comparison, T5-v1.1 is pre-trained on raw texts in a fully unsupervised manner, so the comparison for few-shot QA will be fair. Empirically speaking, in preliminary experiments, we found that

- Vanilla T5-v1.0 always outperformed the continually pre-trained T5-v1.0 significantly under the few-shot setting.

- Vanilla T5-v1.1 outperformed vanilla T5-v1.0 on most adopted datasets under the full-set

fine-tuning settings, which might be the main cause for the higher performance of our baselines compared with the results in Tan et al. (2023).

**Dataset Pre-processing**   To fit the T5 fine-tuning process into one single GPU, we truncate the lengths of the contexts to 1,500 chars for all downstream datasets under single-context and few-shot settings. This may boost performance since slightly shorter contexts make the modeling of dependencies a little easier.

**Hyper-parameter**   The original TempReason paper simply chose a set of hyper-parameters for all experiments. In comparison, we adopted hyper-parameter search in all downstream experiments to avoid biases resulting from hyper-parameter choices. This may boost the performance of our implementation.

**TimeQA-Easy**

Context : *Dorothy Hansine Andersen (May 15, 1901 – March 3, 1963) was an American pathologist and pediatrician who was the first person to identify cystic fibrosis and the first American physician to describe the disease. In 2001 she was inducted into the National Women's Hall of Fame. Early life. Dorothy Hansine Andersen was born in Asheville, North Carolina on May 15, 1901. In 1914 her father, Hans Peter Andersen, died and she took the full responsibility for caring for her invalid mother. Andersen's mother died in 1920 and after they had moved to St. Johnsbury, Vermont. In 1922 Andersen graduated with a Bachelor of Arts in zoology and chemistry from Mount Holyoke College. Later, she went on to attend Johns Hopkins School of Medicine which is where she first began to perform research under Florence Rena Sabin. Andersen's first two research papers were on the lymphatic and blood vessels in the reproductive organs of female pigs. Both of these papers were published in Contributions to Embryology. Once she graduated from Johns Hopkins, Andersen served as a teaching assistant in anatomy at the Rochester School of Medicine. A year later she became an intern for surgery at the Strong Memorial Hospital in Rochester, New York. After completing her internship year, Andersen was denied a residency in general surgery at the hospital because she was a woman. This drove Andersen to focus on her research instead and in 1929, she began working at Columbia University*

Question : *Dorothy Hansine Andersen went to which school from 1921 to 1922?*

Answer : *Mount Holyoke College*

**TimeQA-Hard**

Context : *Mei Li Vos Mei Li Vos (born 31 March 1970 ) is a Dutch politician, former trade unionist and editorialist. A member of the Labour Party ( PvdA ), she was a member of the House of Representatives from 1 March 2007 to 17 June 2010 and again from 20 September 2012 until 23 March 2017. She has been a member of the Senate since 11 June 2019. Early life. Mei Li Vos was born on 31 March 1970 in Eindhoven in the Netherlands . Vos comes from a family with five brothers. Her mother was Chinese Indonesian, established in the Dutch East Indies. During her youth her family lived in a Christian commune in Veldhoven. Between 1982 and 1988 she attended the vwo at the Christian Lyceum in Arnhem. Academic career. Between 1988 and 1989 she studied General Social Science at the University of Utrecht. Vos studied political science at the University of Amsterdam. Between 1994 and 2002 she worked as teacher and researcher at the political science department of the University of Amsterdam. In 1994-1995 she also worked part-time as manager of the Maarten Altena Ensemble. From 1995 and 2000 she worked on her thesis, which concerned the relationship between the Netherlands and Indonesia. She specifically researched what the effect was of Indonesia refusing development cooperation of the*

Question : *What was the position of Mei Li Vos between Feb 2016 and Dec 2016?*

Answer : *member of the House of Representatives*

Table 8: Examples from TimeQA-Hard and -Easy. We underline those context sentence(s) that contain the necessary information to answer the question.

| **TempReason-ReasonQA-L2** |
|---|

Context : *Paul Dubreil studied at Lycée Saint-Louis from Jan 1921 to Jan 1923. Paul Dubreil worked for Université de Nancy from Jan 1933 to Jan 1946. Paul Dubreil studied at École normale supérieure (Paris) from Jan 1923 to Jan 1926. Paul Dubreil worked for University of Paris from Jan 1946 to Jan 1975. Paul Dubreil worked for University of Lille from Jan 1931 to Jan 1933. Paul Dubreil studied at University of Hamburg from Jan 1929 to Jan 1930.*

Question : *Where was Paul Dubreil educated in May 1921?*

Answer : *Lycée Saint-Louis*

| **TempReason-OBQA-L2** |
|---|

Context : *Benoit DoraisBenoit Dorais is a city councillor from Montreal, Quebec, Canada. He has served as the borough mayor of Le Sud-Ouest since 2009. From his first election to 2013, Dorais was a member of Vision Montreal, before joining Coalition Montréal in 2013 and Projet Montréal prior to the 2017 municipal election. Dorais was born and raised in the Saint-Henri neighbourhood. Prior to his election as city councillor, Dorais served as a political staff member to former Bloc Québécois MP Thierry St-Cyr. He has also served as a commissioner with the Commission scolaire de Montréal since 2007. He holds a university degree in philosophy and social ethics. Following Marcel Côté's death on May 26, 2014, Dorais became leader of Coalition Montreal. In 2017 he resigned that role to sit as an independent, joining Projet Montréal soon after, prior to the 2017 elections.Dorais chaired the City of Montreal committee on social development and Montreal diversity. Following the 2017 election, Mayor Valérie Plante named him chair of the Montreal Executive Committee, with responsibility for finances, human resources, and legal affairs.*

Question : *Which political party did Benoit Dorais belong to in Aug 2016?*

Answer : *Coalition Montréal*

| **TempReason-ReasonQA-L3** |
|---|

Context : *Clifford Truesdell studied at California Institute of Technology from Jan 1938 to Jan 1942. Clifford Truesdell worked for Naval Ordnance Laboratory from Jan 1946 to Jan 1948. Clifford Truesdell worked for Johns Hopkins University from Jan 1961 to Jan 1989. Clifford Truesdell worked for Massachusetts Institute of Technology from Jan 1944 to Jan 1946. Clifford Truesdell worked for the United States Naval Research Laboratory from Jan 1948 to Jan 1950. Clifford Truesdell worked for Brown University from Jan 1942 to Jan 1943. Clifford Truesdell worked for Indiana University Bloomington from Jan 1950 to Jan 1961. Clifford Truesdell worked for University of Michigan from Jan 1943 to Jan 1944.*

Question : *Which employer did Clifford Truesdell work for after Indiana University Bloomington?*

Answer : *Johns Hopkins University*

| **TempReason-OBQA-L3** |
|---|

Context : *Piotr WilczekPiotr Antoni Wilczek (born April 26, 1962 in Chorzów) is a Polish intellectual historian, a specialist in comparative literature and a literary translator, who serves as the Ambassador of Poland to the United States.Piotr Wilczek graduated from the University of Silesia in Katowice, Poland (1986) where he received his Ph.D. (1992) and Habilitation (2001) degrees. In 2006 he was nominated professor of the humanities by the President of the Republic of Poland. Doctor of Humane Letters (honoris causa) of Cleveland State University. He was an assistant and associate professor at the University of Silesia (1986–2008), where he also served as Dean of the Faculty of Languages (2002–2008). Since 2008 he has been a tenured full professor at the Faculty of „Artes Liberales", University of Warsaw and until 2016 served there as Head of the Collegium Artes Liberales (College of Liberal Arts and Sciences) and Head of the Centre for the Study of the Reformation and Intellectual Culture in Early Modern Europe.He did his postgraduate work in intellectual history and Neo-Latin Studies at the Universities of Oxford (St Anne's College, 1988) and Łódź, Poland (1989). He was a visiting translator at The British Centre for Literary Translation, University o*

Question : *Which employer did Piotr Wilczek work for before University of Warsaw?*

Answer : *University of Silesia*

Table 9: Examples from TempReason, including ReasonQA Level-2 & Level-3 and Open-domain QA (OBQA) Level-2 & Level-3. We underline those context sentence(s) that contain the necessary information to answer the question.

**SituatedQA**

Context : *USA on her first try, then went on to participate in Miss USA 1997 at Shreveport, Louisiana on February 5, 1997, where she was crowned the winner by outgoing titleholder Ali Landry of Louisiana. Lee represented the United States in the Miss Universe 1997 pageant in Miami Beach, Florida. On May 16, 1997, she won the crown at 26 years and 128 days, became the oldest Miss Universe to win. Lee and Al Masini along with funding from the state were forces behind the Miss Universe 1998 pageant being held in her home state of Hawaii, in Honolulu, for the first time. Immediately prior to winning the Miss USA and Miss Universe crowns Lee Miss Universe, after winning Miss USA. Six titleholders represented the United States or Hawaii at major international pageants, with Miss Hawaii USA 1997, Brook Lee, winning the title of Miss USA 1997 and capturing the crown of Miss Universe 1997, one of only 8 Miss USA winners to become Miss Universe in the history of the pageant. Hawaii holds a record of 24 placements at Miss USA.*

Question : *When was the last time USA won miss universe as of July 22, 1998?*

Answer : *May 16, 1997*

Table 10: Examples from SituatedQA. We underline those context sentence(s) that contain the necessary information to answer the question.