# OpenReview forum: "Once Upon a ${\it Time}$ in ${\it Graph}$: Relative-Time Pretraining for Complex Temporal Reasoning"
_EMNLP/2023/Conference — EMNLP 2023 Main_

### Official Review · Reviewer_r1Nz · 2023-08-04

**Soundness:** 4

**Excitement:**

3: Ambivalent: It has merits (e.g., it reports state-of-the-art results, the idea is nice), but there are key weaknesses (e.g., it describes incremental work), and it can significantly benefit from another round of revision. However, I won't object to accepting it if my co-reviewers champion it.

**Paper Topic And Main Contributions:**

To better capture the temporal dependencies, this paper proposes a new pretraining task, REMEMO, which predicts the temporal relation between any two sentences in a document. Specifically, REMEMO first identifies the time token in the documents and builds a normalized time span for every sentences, like [2003-01-01, 2004-01-01). Then, by comparing the time spans of two sentences, the temporal relation between two sentences could be classified as Earlier, Later and Contemporary. During pretraining, REMEMO requires the model to predict the temporal relation based on the sentence representations. This paper apply REMEMO to T5 and shows greaat improvements on several temporal question answering datasets.

**Questions For The Authors:**

1. In few-shot settings, REMEMO performs worse than T5+LM in many tasks, like OBQA L3. This is strange for me, because REMEMO has been pretrained on the domain-specific task and I think it should performe much better than T5 in fewshot settings.
2. For those sentences that don't contain time token, how to determine the temporal relation between them and other sentences?

**Reasons To Accept:**

1. The proposed pretraining task, REMEMO, is simple yet effective. This method might be useful to alleviate the problem that LLMs cannot handle temporal reasoning well.

2. The ablation study with different context length shows the REMEMO could alleviate the degration brought by long input. This result shows the REMEMO has the ability to capture the most important temporal dependency and ignore the irrelevant information.

**Reasons To Reject:**

1. I don't understand why the authors use ``Graph`` to explain the proposed method. I think the motivation could be very easy to understand, that is identifying the temporal relation between two sentences during pretraining. But adding ``Graph`` makes me confused when reading the paper.

2. The comparison with previous works is not sufficient. In the experiment, the authors only report their own baselines but don't include any previous efforts, like the  TempT5 proposed by Tan[1]. Besides, the perfromance of T5 in TempReason seems to be much higher than what is reported in the original paper[1]. Any explanation for it?

3. The authors hope to model the temporal dependencies facts, but they seem to ignore a line of works that directly solve this problem, that is event temporal relation classification[2-6]. Can we first finetune LMs in such task and then apply them to solve temporal QA? At least, I think the authors should discuss them in related work.

[1] Qingyu Tan, et al. Towards benchmarking and improving the temporal reasoning capability of large language models. ACL 2023.

[2] Taylor Cassidy et al. An annotation framework for dense event ordering. ACL 2014.

[3] Qiang Ning et al. A multi-axis annotation scheme for event temporal relations. ACL 2018.

[4] Puneet Mathur, et al. TIMERS: Document-level temporal relation extraction. ACl 2021.

[5] Shuaicheng Zhang, et al. Extracting temporal event relation with syntactic-guided temporal graph transformer. NAACL 2022.

[6] Quzhe Huang, et al. More than Classification: A Unified Framework for Event Temporal Relation Extraction. ACL 2023.

**Reproducibility:**

4: Could mostly reproduce the results, but there may be some variation because of sample variance or minor variations in their interpretation of the protocol or method.

**Reviewer Confidence:**

4: Quite sure. I tried to check the important points carefully. It's unlikely, though conceivable, that I missed something that should affect my ratings.

---

> ### Author Rebuttal · Authors · 2023-08-29
>
> We would like to thank Reviewer r1Nz for the insightful comments and suggestions that have helped identify our work's strengths and potential improvements.
>
> ## 1. Why use `Graph` to explain the proposed method. The motivation could be very easy to understand, that is identifying the temporal relation between two sentences during pretraining.
>
> We use `graph` instead of temporal relation to explain our method because it is not mere temporal relation itself works. We show that REMEMO remains highly effective as long as “high-order” temporal relations exist among sentences during pre-training. This is rather similar to the scenario where there exist paths (either one-edge or multi-edge) from any node to another in a graph. This suggests that the internal dynamics during REMEMO pre-training may go beyond simple “one-order” time relation classification. Therefore, it might be better to explain REMEMO from a graph view.
>
> ### 1.1 Using Only Time Relation Classification
>
> When initiating this work, our very first idea was to construct BERT-like pre-training instances, i.e., ```[CLS] Sentence-A [SEP] Sentence-B [SEP]```. The model is required to predict the time relation of Sentence-A to Sentence-B during pre-training. However, preliminary experiments showed that this method did not offer any substantial improvement over the baseline. We thus tried out a few modifications and found out that **including multiple sentences in one training instance** works well.
>
> ### 1.2 Intuitive Explanation
>
> Our time-relation classification (TRC) objective aims to classify the time relation between any two sentences, which can be roughly seen as "first-order" time relation classification. However, the input of our REMEMO usually contains far more than two sentences (8~9 on average). Such a large number of sentences induces complex sentence-level dependencies and there may exist "high-order" sentence-to-sentence temporal connections that will affect the "first-order" relation classification (i.e., the TRC objective).
>
> As an intuitive example, suppose we have 4 sentences, $\{S_1, S_2, S_3, S_4\}$, in one instance. The prediction from $S_1$ to $S_2$ may be influenced by the predictions (i) from $S_1$ to $S_3$; (ii) from $S_2$ to $S_3$; (iii) ... et al. If the model tends to predict that (i) $S_1$ is earlier than $S_3$ and (ii) $S_2$ is later than $S_3$, it is thus likely that the model would predict $S_1$ is earlier than $S_2$. Here, $S_3$ serves as a bridge node to construct a "second-order" time relation between $S_1$ and $S_2$. Similar ideas apply to higher-order relations. Overall, though the final prediction target is a simple "first-order" time relation, all these 8~9 sentences may have complex interactions with each other either directly or indirectly across many Transformer layers.
>
> ### 1.3 Empirical Analysis
>
> The above intuition may be supported by Table 6 of our paper. We briefly list the results below. `# Paths` refers to the number of paths from any node to another. In other words, `# Paths ≥ 1 for most nodes` implies that there exist “high-order” connections among most sentences.
>
> | Method | # Paths | Averaged F1 | Averaged EM |
> | --- | --- | --- | --- |
> | T5+LM | # Paths $=0$ | 62.71 | 52.66 |
> | REMEMO with $n$ TRC Labels | # Paths $\geq 1$ for most nodes if the edges distribute randomly | 67.67 | 58.77 |
> | REMEMO with $\frac{n(n-1)}{\log n}$ TRC Labels | # Paths $\geq 1$ for almost all nodes if the edges distribute randomly | 68.09 | 58.89 |
> | REMEMO with $n(n-1)$ TRC Labels | # Paths $=13,699$ for all nodes (suppose $n=8$) | 68.29 | 59.33 |
>
> As shown in the table above, the three respective REMEMO models with $n$, $\frac{n(n-1)}{\log n}$ and $n(n-1)$ edges show similar performances and all significantly outperform T5+LM (i.e., with $0$ edges). This indicates that REMEMO remains effective as long as “high-order” temporal connections exist, which is similar to a graph where nodes are connected through multiple edges. We thus explain our method from the `Graph` view.
>
> ## 2. The performance of T5 in TempReason seems to be much higher than what is reported in the original paper. The comparison with previous works is not sufficient.
>
> ## TL;DR:
>
> 1. The higher performances compared with the results reported from the TempReason paper are caused by a few differences between our implementation and the implementation of the authors of TempReason.
> 2. We add two tables to report comparisons with other existing methods, including (i) TempT5 on the TempReason dataset; and (ii) DPR and human evaluation on the SituatedQA dataset. Results show that REMEMO remains the best of all.
>
> ## Detailed Author Response:
>
> ### 2.1 The performance of T5 in TempReason seems to be much higher than what is reported in the original paper [1].
>
> We appreciate the reviewer for pointing out this issue. The different performances result from a few differences between our implementation and the implementation of the authors of TempReason.
>
> - **T5 Versions**: As mentioned in Section 4.1.2 of our paper, we adopted T5-v1.1 for all experiments, while the original TempReason papers adopted T5-v1.0. We did not adopt T5-v1.0 because its pre-training corpora included downstream datasets, such as SQuAD, which makes the evaluation of few-shot QA unfair. In comparison, T5-v1.1 is pre-trained on raw texts in a fully unsupervised manner, so the comparison for few-shot QA will be fair. Empirically speaking, in preliminary experiments, we found that
>     - Vanilla T5-v1.0 always outperformed the continually pre-trained T5-v1.0 significantly under the few-shot setting.
>     - Vanilla T5-v1.1 outperformed vanilla T5-v1.0 for roughly 5 to 10 points in terms of F1-scores on most adopted datasets under the full-set fine-tuning settings. We assume that this is the main cause for the higher performance of our baselines compared with the results in the original TempReason paper.
> - **Dataset Pre-processing**: To fit the T5 fine-tuning process into one single GPU, we truncate the lengths of the contexts to 1,500 chars for all downstream datasets under single-context and few-shot settings. This may boost performance since slightly shorter contexts make the modeling of dependencies a little easier. We mentioned this in Line 423 of our paper.
> - **Hyper-parameter**: The original TempReason paper simply chose a set of hyper-parameters for all experiments. In comparison, we adopted hyper-parameter search in all downstream experiments to avoid biases resulting from hyper-parameter choices. This may boost the performance of our implementation. We listed detailed hyper-parameters in Appendix A.2 of our paper.
>
> We will highlight the above implementation differences in the next version of our paper.
>
> ### 2.2 The comparison with previous works is not sufficient.
>
> ### TL;DR:
>
> 1. There are only a small number of existing methods focusing on temporal question answering. Among them, TempT5 is the only proper one for comparison. It was proposed in the TempReason dataset paper.
> 2. TempT5 improves the performance of T5+LM for +$0.54\%$ F1-scores, but does not improve over our best model (REMEMO+SFT) and even slightly hurts its performance.
> 3. Our best model remains the best of all.
>
> ### Detailed Author Response:
>
> Temporal question answering (TQA), as a specific task, has garnered limited attention in earlier years, resulting in few existing methods for comparison. As far as we know, all existing non-GPT methods that conducted experiments on the three datasets we adopted come from their corresponding original dataset papers.
>
> - **TimeQA**: Their original dataset paper adopts Fusion-in-Decoder (FiD), for which we have implemented the experiments and reported the results in Table 4 of our paper.
> - **TempReason**: Their dataset paper proposed TempT5, a reinforcement learning (RL) method that further improves supervised fine-tuned (SFT) T5 models. It is a meaningful comparison for our experiments. We thus conducted additional experiments using this RL method and include the results in Table 1 below.
> - **SituatedQA**: Their dataset paper adopts Dense Passage Retrieval (DPR) [1] as the baseline. Apart from the retrieval component, DPR is no more than a simple BERT-based extractive QA baseline. We list the DPR results reported by the SituatedQA paper in Table 2 below.
>
> **Table 1**: F1-scores on TempReason. All experiments are conducted using T5-v1.1-base. "SFT+RL" refers to TempT5.
>
> | Pre-training Method | Fine-tuning Method | Temp-L2 | OBQA-L2 | Temp-L3 | OBQA-L3 | Average |
> | --- | --- | --- | --- | --- | --- | --- |
> | T5 | SFT | 90.62 | 44.99 | 59.56 | 41.76 | 59.23 |
> | T5 | SFT+RL | 90.61 | 44.64 | 59.88 | 41.42 | 59.14 |
> | T5+LM | SFT | 94.64 | 49.03 | 62.71 | 43.02 | 62.35 |
> | T5+LM | SFT+RL | 95.05 | 49.60 | 63.92 | 43.00 | 62.89 |
> | REMEMO | SFT | 97.67 | 51.58 | 91.17 | 44.91 | 71.33 |
> | REMEMO | SFT+RL | 97.61 | 51.14 | 91.28 | 44.70 | 71.18 |
>
> As can be seen from Table 1 above, SFT+RL (i.e., TempT5) only improves the averaged performance of T5+LM but does not work for all other models. Specifically, REMEMO+SFT (our best model) already obtains the best results among all of them; and further applying RL to our REMEMO+SFT slightly hurts its performance. This may be caused by the fact that all our T5+SFT checkpoints (including baselines and REMEMO) are obtained through extensive hyperparameter search so little room is left for further improvement through further dataset-specific RL training.
>
> **Table 2**: Results on SituatedQA. DPR and Human* results are from the original SituatedQA dataset paper. Human* results were from human evaluation conducted on 50 randomly-sampled instances.
>
> | Method | F1 | EM |
> | --- | --- | --- |
> | DPR | N/A | 23.00 |
> | REMEMO-base | 55.06 | 48.12 |
> | REMEMO-large | 56.06 | 49.12 |
> | Human* | N/A | 57.00 |
>
> > Notes:
> >
> > 1. Because TempT5 requires the dataset to have negative answers in addition to the correct answer, we can only conduct TempT5 experiments on TempReason, which is the only dataset that contains negative answer candidates.
> > 2. The original TempReason paper adopted T5-v1.0 for experiments, while we adopted T5-v1.1 in all our experiments. We did not adopt T5-v1.0 because its pre-training corpora included downstream datasets, such as SQuAD, which will make the evaluation of few-shot QA unfair. In comparison, T5-v1.1 is pre-trained on raw texts in a fully-unsupervised manner.
> > 3. The original SituatedQA paper only reported exact match (EM) scores.
> > 4. For the results in Table 1 above, the default setting in TempReason's released code is 'num_layers_unfrozen'=6 and 'max_len'= 768. We additionally tried 'num_layers_unfrozen'$\in$ {-1, 6, 2, 1} and 'max_len' $\in$ {768, 512}. We then take the models that obtain the best validation F1-scores for testing.
>
> ## 3. Related to event temporal relation classification
>
> We agree with the reviewer our work is closely related to previous work that focused on event temporal relation classification (ETRC). TimeBank, on which we trained the time-identification model, belongs to this line of work. We do appreciate the efforts that were made by this line of work. Below we discuss some similarities and differences between our work and previous ETRC works. We will add these discussions and related references ([1], [2], [3], [4], [5]) in the next version of our paper.
>
> ### 3.1 Focus on temporal relations but on different granularities
>
> The main concept shared by our work and previous ETRC works is the temporal relation between two events. However, some task differences exist. Our work focuses on temporal question answering (TQA), in which most clues in existing datasets are **explicit and coarse-grained time expressions**, such as "*June 2012*", "*18th century*" and "*1980s*". In contrast, the line of ETRC works focused on time information that is **more implicit, fine-grained, and nuanced**. Below we show some contrasting examples.
>
> **Temporal Question Answering (Clues to answer the question are bold.)**
>
> - Example-1 (from TimeQA-hard):
>
>     Context: Mei Li Vos Mei Li Vos (born 31 March 1970 ) is a Dutch politician, former trade unionist and editorialist. A member of the Labour Party ( PvdA ), **she was a member of the House of Representatives from 1 March 2007 to 17 June 2010 and again from 20 September 2012 until 23 March 2017.** She has been a member of the Senate since 11 June 2019.
>
>     Question: What was the position of Mei Li Vos between Feb 2016 and Dec 2016?
>
>     Answer: member of the House of Representatives.
>
>
> - Example-2 (from SituatedQA):
>
>     Context: Ray Allen is widely considered to be one of the greatest three-point shooters of all-time, and **he held the record for most three-pointers made in a career from 2011 until 2021**, when he was surpassed by Stephen Curry. ........... During a game against the Los Angeles Lakers **on February 10, 2011, Allen became the all-time NBA leader in total three-pointers made (2,562), surpassing Reggie Miller's record of 2,560**.
>
>     Question-1: Who made the most three point shots in total in the NBA as of 2012?
>
>     Answer-1: Ray Allen.
>
>     Question-2: Who made the most three point shots in total in the NBA as of 2010?
>
>     Answer-2: Reggie Miller.
>
>
> **Event Temporal Relation Classification**
>
> - Example-1 (from [1]):
>
>     Police **confirmed** **Friday** that the body **found** along a highway in San Juan **belonged** to Jorge Hernandez.
>
>     TimeBank-Dense Labels: *belonged* before *confirmed*; *belonged* before *found*; *found* before *confirmed*; *belonged* before *Friday*; *confirmed* is-included-in *Friday*; *found* is-included-in *Friday*.
>
> - Example-2 (from [2]):
>
>     At one point , when it **became** clear controllers could not contact the plane, someone **said** a prayer.
>
>     MATRES Label: *became* before *said*.
>
> - Example-3 (from [2]):
>
>     The US is bolstering its military presence in the gulf, as President Clinton **discussed** the Iraq crisis with the one ally who has **backed** his threat of force, British prime minister Tony Blair.
>
>     MATRES Label: *discussed* after *backed*.
>
>
> ### 3.2 Modeling
>
> Several methods on ETRC ([3], [4]) adopted graph neural networks to model ETRC from a graph view. We also propose a graph-like view over our extracted temporal relations from raw corpora and incorporate this view into our modeling process. However, we make efforts to incorporate our proposed graph view into large-scale pre-training and empirically demonstrate the effectiveness of such pre-training.
>
> ### 3.3 Can we first finetune LMs in event temporal relation classification and then apply them to solve temporal QA?
>
> We did not try the exact method above, but we used to conduct experiments under a similar setting which may provide some information. We constructed time relation labels on the “context” of temporal question-answering (TQA) datasets. We tried to directly fine-tune a vanilla T5 with two joint objectives: the answer generation objective and the time relation classification (TRC) objective. Below are the F1 scores. Note that these results were early trials and **were conducted in a different setting from those of the reported results in our paper, making them not directly comparable**.
>
> | Method | TimeQA-Easy | TimeQA-Hard | ReasonQA-L2 | OBQA-L2 | ReasonQA-L3 | OBQA-L3 |
> | --- | --- | --- | --- | --- | --- | --- |
> | T5-base | 65.77 | 61.69 | 85.10 | 37.04 | 60.89 | 31.89 |
> | T5-base + TRC fine-tuning | 61.13 | 54.37 | 86.56 | 34.82 | 74.08 | 32.53 |
>
> The table above shows that directly applying TRC in the fine-tuning stage yields mixed results. Though TRC shows a large improvement on ReasonQA-L3, it also gives significant decreases on TimeQA-Easy, TimeQA-Hard, and OBQA-L2. Based on the results above, we believe pre-training is a necessary step for adapting temporal relation classification to solve temporal QA.
>
> ## 4. In few-shot settings, REMEMO performs worse than T5+LM in many tasks, like OBQA L3. This is strange because REMEMO has been pre-trained on the domain-specific task and it should perform much better than T5 in few-shot settings.
>
> This may be caused by
>
> 1. Even though REMEMO is pre-trained with one additional objective: time relation classification (TRC), this classification objective is not directly aligned with the answer generation objective of downstream temporal QA tasks.
> 2. T5+LM is only pre-trained with seq-to-seq generation and does not inherit "noise" from TRC, so it might be more directly applicable to QA, whose format is also seq-to-seq generation.
> 3. With slightly more instances (i.e., increasing from 16-shot to 64-shot), REMEMO starts to outperform T5+LM on most datasets. This may suggest that REMEMO requires a reasonable but not large number of instances to rectify its behavior into answer generation.
>
> ## 5. For those sentences that don't contain time token, how to determine the temporal relation between them and other sentences?
>
> We excluded such sentences when constructing time relation classification (TRC) labels.
>
> ## References:
>
> - [1] Taylor Cassidy et al. An annotation framework for dense event ordering. ACL 2014.
> - [2] Qiang Ning et al. A multi-axis annotation scheme for event temporal relations. ACL 2018.
> - [3] Puneet Mathur et al. TIMERS: Document-level Temporal Relation Extraction. ACL 2021.
> - [4] Shuaicheng Zhang et al. Extracting Temporal Event Relation with Syntax-guided Graph Transformer. NAACL 2022.
> - [5] Quzhe Huang, et al. More than Classification: A Unified Framework for Event Temporal Relation Extraction. ACL 2023.

---

### Official Review · Reviewer_ZZGY · 2023-08-04

**Soundness:** 3

**Excitement:**

4: Strong: This paper deepens the understanding of some phenomenon or lowers the barriers to an existing research direction.

**Paper Topic And Main Contributions:**

This paper proposes REMEMO, aiming to make use of time information by modeling time relations among sentences. Experimental results demonstrate the effectiveness of REMEMO.

**Questions For The Authors:**

How about the efficiency of REMEMO, e.g., the training time on each dataset, which can be used to evaluate whether the proposed method can be used for scale up over real-world application?

**Reasons To Accept:**

1. A TRC pre-training objective is designed for complex temporal reasoning.
2. The performance improvements are remarkable.

**Reasons To Reject:**

1. The baseline model seems to be less.
2. The process of extracting relation (i.e., Later, Earlier, Contemp) is not illustrated explicitly.

**Reproducibility:**

4: Could mostly reproduce the results, but there may be some variation because of sample variance or minor variations in their interpretation of the protocol or method.

**Reviewer Confidence:**

2: Willing to defend my evaluation, but it is fairly likely that I missed some details, didn't understand some central points, or can't be sure about the novelty of the work.

---

> ### Author Rebuttal · Authors · 2023-08-29
>
> We would like to thank Reviewer ZZGY for the insightful comments and suggestions that have helped identify our work's strengths and potential improvements.
>
> ## 1. The baseline model seems to be less.
>
> ### TL;DR:
>
> 1. There are only a small number of existing methods focusing on temporal question answering. Among them, TempT5 is the only proper one for comparison. It was proposed in the TempReason dataset paper.
> 2. TempT5 improves the performance of T5+LM for +$0.54\%$ F1-scores, but does not improve over our best model (REMEMO+SFT) and even slightly hurts its performance.
> 3. Our best model remains the best of all.
>
> ### Detailed Author Response:
>
> Temporal question answering (TQA), as a specific task, has garnered limited attention in earlier years, resulting in few existing methods for comparison. As far as we know, all existing non-GPT methods that conducted experiments on the three datasets we adopted come from their corresponding original dataset papers.
>
> - **TimeQA**: Their original dataset paper adopts Fusion-in-Decoder (FiD), for which we have implemented the experiments and reported the results in Table 4 of our paper.
> - **TempReason**: Their dataset paper proposed TempT5, a reinforcement learning (RL) method that further improves supervised fine-tuned (SFT) T5 models. It is a meaningful comparison for our experiments. We thus conducted additional experiments using this RL method and include the results in Table 1 below.
> - **SituatedQA**: Their dataset paper adopts Dense Passage Retrieval (DPR) [1] as the baseline. Apart from the retrieval component, DPR is no more than a simple BERT-based extractive QA baseline. We list the DPR results reported by the SituatedQA paper in Table 2 below.
>
> **Table 1**: F1-scores on TempReason. All experiments are conducted using T5-v1.1-base. "SFT+RL" refers to TempT5.
>
> | Pre-training Method | Fine-tuning Method | Temp-L2 | OBQA-L2 | Temp-L3 | OBQA-L3 | Average |
> | --- | --- | --- | --- | --- | --- | --- |
> | T5 | SFT | 90.62 | 44.99 | 59.56 | 41.76 | 59.23 |
> | T5 | SFT+RL | 90.61 | 44.64 | 59.88 | 41.42 | 59.14 |
> | T5+LM | SFT | 94.64 | 49.03 | 62.71 | 43.02 | 62.35 |
> | T5+LM | SFT+RL | 95.05 | 49.60 | 63.92 | 43.00 | 62.89 |
> | REMEMO | SFT | 97.67 | 51.58 | 91.17 | 44.91 | 71.33 |
> | REMEMO | SFT+RL | 97.61 | 51.14 | 91.28 | 44.70 | 71.18 |
>
> As can be seen from Table 1 above, SFT+RL (i.e., TempT5) only improves the averaged performance of T5+LM but does not work for all other models. Specifically, REMEMO+SFT (our best model) already obtains the best results among all of them; and further applying RL to our REMEMO+SFT slightly hurts its performance. This may be caused by the fact that all our T5+SFT checkpoints (including baselines and REMEMO) are obtained through extensive hyperparameter search so little room is left for further improvement through further dataset-specific RL training.
>
> **Table 2**: Results on SituatedQA. DPR and Human* results are from the original SituatedQA dataset paper. Human* results were from human evaluation conducted on 50 randomly-sampled instances.
>
> | Method | F1 | EM |
> | --- | --- | --- |
> | DPR | N/A | 23.00 |
> | REMEMO-base | 55.06 | 48.12 |
> | REMEMO-large | 56.06 | 49.12 |
> | Human* | N/A | 57.00 |
>
> > Notes:
> >
> > 1. Because TempT5 requires the dataset to have negative answers in addition to the correct answer, we can only conduct TempT5 experiments on TempReason, which is the only dataset that contains negative answer candidates.
> > 2. The original TempReason paper adopted T5-v1.0 for experiments, while we adopted T5-v1.1 in all our experiments. We did not adopt T5-v1.0 because its pre-training corpora included downstream datasets, such as SQuAD, which will make the evaluation of few-shot QA unfair. In comparison, T5-v1.1 is pre-trained on raw texts in a fully-unsupervised manner.
> > 3. The original SituatedQA paper only reported exact match (EM) scores.
> > 4. For the results in Table 1 above, the default setting in TempReason's released code is 'num_layers_unfrozen'=6 and 'max_len'= 768. We additionally tried 'num_layers_unfrozen'$\in$ {-1, 6, 2, 1} and 'max_len' $\in$ {768, 512}. We then take the models that obtain the best validation F1-scores for testing.
>
> ## 2. The process of extracting relation (i.e., Later, Earlier, Contemp) is not illustrated explicitly.
>
> We thank the reviewer for pointing out this issue. We will add more detailed explanations and input-output pair examples of our pre-processing pipeline in the next version of our paper.
>
> To help better illustrate the details of our pre-processing pipeline, we plan to do the following:
>
> - We will add detailed explanations of the time-identification phase and the time-normalization phase to the appendix for the next version of our paper.
>     - The time-identification phase is to fine-tune a RoBERTa model using TimeBank v1.2 Dataset. We will add several cases on (i) how this phrase works; and (ii) what the input and output could be.
>     - The time-normalization phase is to normalize the time-expressions that were extracted from the previous phase using a rule-based script. We have provided the link to the script in Line #207 of our paper and shown several input-output examples in Table 2 of our paper.
> - We will release the code for all experiments, including the pre-processing pipeline and the time-tagging fine-tuned RoBERTa checkpoint after the anonymous period, so that all details will be clear then.
>
> ## 3. How about the efficiency of REMEMO, e.g., the training time on each dataset, which can be used to evaluate whether the proposed method can be used for scale up over real-world application?
>
> ### TL;DR:
>
> Fine-tuning normally takes 1~3 hours using a single GPU.
>
> ### Detailed Author Response:
>
> The efficiency of REMEMO is two-fold:
>
> - **Pre-training Cost**: We reported the pre-training cost in Section 4.1.2 of our paper.
> - **Fine-tuning Cost**: The fine-tuning cost of REMEMO is the same as a usual T5 model. Specifically, we fine-tune each model for 10 epochs on each dataset. The specific training time depends on the size of the corresponding dataset. Below are a few examples:
>     - Fine-tuning a base-size model on TempReason-ReasonQA-L2 takes about 3 hours using a single V100-32G GPU.
>     - Fine-tuning a base-size model on TimeQA-hard takes about 2.5 hours using a single V100-32G GPU.
>     - Fine-tuning a large-size model on TimeQA-hard takes about 2.5 hours using a single A100-80G GPU.
>     - Fine-tuning a large-size model on SituatedQA takes about 1.5 hours using a single A100-80G GPU.
>
> We will add a discussion paragraph regarding the time cost of REMEMO to Section *Limitations* in the next version of our paper.

---

### Official Review · Reviewer_bSAa · 2023-08-10

**Soundness:** 4

**Excitement:**

4: Strong: This paper deepens the understanding of some phenomenon or lowers the barriers to an existing research direction.

**Paper Topic And Main Contributions:**

This paper addresses the challenge of enhancing the understanding and reasoning capabilities of pre-trained language models in dealing with temporal contexts in text.The main problem at hand is the difficulty of capturing and utilizing temporal dependencies between pieces of knowledge in various downstream tasks. While existing approaches mainly focus on connecting text snippets directly to timestamps, they often fall short when more complex reasoning over temporal relationships is required.

The main contributions of the paper can be summarized as follows:

1. REMEMO Framework: The paper introduces the REMEMO framework, which goes beyond simple knowledge-time association and instead focuses on capturing complex temporal dependencies between different pieces of knowledge. This framework utilizes a graph-based approach to represent and model these dependencies, offering a more comprehensive understanding of temporal contexts in text.

2. Improved Performance: Experimental results demonstrate that REMEMO outperforms the baseline model T5 on multiple temporal question answering datasets under various settings. This indicates that REMEMO's approach of explicitly modeling time relations between sentences leads to more accurate and effective reasoning over temporal contexts.

3. Long-Range Temporal Dependencies: The paper provides further analysis indicating that REMEMO excels at modeling long-range and complex temporal dependencies. This capability is crucial for a wide range of applications where understanding distant relationships in time is essential.

**Reasons To Accept:**

1. Comprehensive Temporal Understanding: REMEMO goes beyond conventional knowledge-time association methods and offers a more detailed understanding of temporal contexts.

2. Long-Range Dependency Handling: The paper's focus on modeling long-range complex temporal dependencies is a significant contribution. This capability is highly relevant in real-world scenarios where understanding distant relationships in time is crucial for accurate comprehension and reasoning.


Main benefits for the NLP community:
1. The paper contributes to advancing the state-of-the-art in temporal reasoning within NLP. By addressing a critical challenge in understanding temporal contexts, it provides valuable tools and techniques for researchers and practitioners to build more accurate and sophisticated models

2.  Incorporating the REMEMO framework could lead to improved performance of NLP models across various tasks that involve temporal information. This could have a cascading effect on downstream applications, leading to better user experiences and more reliable results.

**Reasons To Reject:**

1. Comparison with Existing Methods: While REMEMO outperforms the baseline T5, a more comprehensive comparison with other state-of-the-art methods could provide a clearer understanding of its relative performance and strengths. Ensuring that the comparison covers a diverse set of temporal reasoning models could add more depth to the evaluation.

2. Impact of Noise and Ambiguity: Temporal contexts in text can sometimes be ambiguous or noisy. The paper could discuss how REMEMO handles cases where temporal information is unclear or conflicting, and whether it has limitations in such scenarios.

**Reproducibility:**

5: Could easily reproduce the results.

**Reviewer Confidence:**

4: Quite sure. I tried to check the important points carefully. It's unlikely, though conceivable, that I missed something that should affect my ratings.

---

> ### Author Rebuttal · Authors · 2023-08-29
>
> We would like to thank Reviewer bSAa for the insightful comments and suggestions that have helped identify our work's strengths and potential improvements.
>
> ## 1. Comparison with Existing Methods
>
> ### TL;DR:
>
> 1. There are only a small number of existing methods focusing on temporal question answering. Among them, TempT5 is the only proper one for comparison. It was proposed in the TempReason dataset paper.
> 2. TempT5 improves the performance of T5+LM for +$0.54\%$ F1-scores, but does not improve over our best model (REMEMO+SFT) and even slightly hurts its performance.
> 3. Our best model remains the best of all.
>
> ### Detailed Author Response:
>
> Temporal question answering (TQA), as a specific task, has garnered limited attention in earlier years, resulting in few existing methods for comparison. As far as we know, all existing non-GPT methods that conducted experiments on the three datasets we adopted come from their corresponding original dataset papers.
>
> - **TimeQA**: Their original dataset paper adopts Fusion-in-Decoder (FiD), for which we have implemented the experiments and reported the results in Table 4 of our paper.
> - **TempReason**: Their dataset paper proposed TempT5, a reinforcement learning (RL) method that further improves supervised fine-tuned (SFT) T5 models. It is a meaningful comparison for our experiments. We thus conducted additional experiments using this RL method and include the results in Table 1 below.
> - **SituatedQA**: Their dataset paper adopts Dense Passage Retrieval (DPR) [1] as the baseline. Apart from the retrieval component, DPR is no more than a simple BERT-based extractive QA baseline. We list the DPR results reported by the SituatedQA paper in Table 2 below.
>
> **Table 1**: F1-scores on TempReason. All experiments are conducted using T5-v1.1-base. "SFT+RL" refers to TempT5.
>
> | Pre-training Method | Fine-tuning Method | Temp-L2 | OBQA-L2 | Temp-L3 | OBQA-L3 | Average |
> | --- | --- | --- | --- | --- | --- | --- |
> | T5 | SFT | 90.62 | 44.99 | 59.56 | 41.76 | 59.23 |
> | T5 | SFT+RL | 90.61 | 44.64 | 59.88 | 41.42 | 59.14 |
> | T5+LM | SFT | 94.64 | 49.03 | 62.71 | 43.02 | 62.35 |
> | T5+LM | SFT+RL | 95.05 | 49.60 | 63.92 | 43.00 | 62.89 |
> | REMEMO | SFT | 97.67 | 51.58 | 91.17 | 44.91 | 71.33 |
> | REMEMO | SFT+RL | 97.61 | 51.14 | 91.28 | 44.70 | 71.18 |
>
> As can be seen from Table 1 above, SFT+RL (i.e., TempT5) only improves the averaged performance of T5+LM but does not work for all other models. Specifically, REMEMO+SFT (our best model) already obtains the best results among all of them; and further applying RL to our REMEMO+SFT slightly hurts its performance. This may be caused by the fact that all our T5+SFT checkpoints (including baselines and REMEMO) are obtained through extensive hyperparameter search so little room is left for further improvement through further dataset-specific RL training.
>
> **Table 2**: Results on SituatedQA. DPR and Human* results are from the original SituatedQA dataset paper. Human* results were from human evaluation conducted on 50 randomly-sampled instances.
>
> | Method | F1 | EM |
> | --- | --- | --- |
> | DPR | N/A | 23.00 |
> | REMEMO-base | 55.06 | 48.12 |
> | REMEMO-large | 56.06 | 49.12 |
> | Human* | N/A | 57.00 |
>
> > Notes:
> >
> > 1. Because TempT5 requires the dataset to have negative answers in addition to the correct answer, we can only conduct TempT5 experiments on TempReason, which is the only dataset that contains negative answer candidates.
> > 2. The original TempReason paper adopted T5-v1.0 for experiments, while we adopted T5-v1.1 in all our experiments. We did not adopt T5-v1.0 because its pre-training corpora included downstream datasets, such as SQuAD, which will make the evaluation of few-shot QA unfair. In comparison, T5-v1.1 is pre-trained on raw texts in a fully-unsupervised manner.
> > 3. The original SituatedQA paper only reported exact match (EM) scores.
> > 4. For the results in Table 1 above, the default setting in TempReason's released code is 'num_layers_unfrozen'=6 and 'max_len'= 768. We additionally tried 'num_layers_unfrozen'$\in$ {-1, 6, 2, 1} and 'max_len' $\in$ {768, 512}. We then take the models that obtain the best validation F1-scores for testing.
>
> ## 2. Noise and Ambiguity of Temporal Contexts in Text
>
> ### TL;DR:
>
> We adopted an aggressive filtering strategy for data pre-processing to avoid noises as much as we could.
>
> ### Detailed Author Response:
>
> We agree with the reviewer that temporal contexts in text can sometimes be ambiguous or noisy. Below we discuss how we handle such noise:
>
> - Generally speaking, we maintained strict filtering during pre-training data pre-processing, i.e., high precision rather than high recall. We adopted such a strategy because even little bits of noise might be highly harmful and the amount of raw pre-training articles is quite sufficient.
> - Specifically:
>     - Unclear temporal information: We regard such sentences as "not containing a valid time-span tag" and do not include them when constructing time-relation classification (TRC) labels.
>     - Conflicting temporal information: As discussed in Section 3.1 of our paper, we merge multiple time spans, supported by the intuition that time expressions that appeared in the same sentence tend to describe events that happened during a similar period of time. Therefore, when multiple mentions are found, they are regarded as a whole. For example,
>         - Victor Hugo's most famous works are the novels *The Hunchback of Notre-Dame* (1831) and *Les Misérables* (1862).
>         - García Márquez started as a journalist and wrote many acclaimed non-fiction works and short stories, but is best known for his novels, such as *One Hundred Years of Solitude* (1967), *Chronicle of a Death Foretold* (1981), and *Love in the Time of Cholera* (1985).
>         - Both sentences above contain more than one time-expressions, but the two mergers of time-spans, $[18310101, 18630101)$ and $[19670101, 19860101)$, are meaningful and distinguishable from each other.
>
> ## References
>
> - [1] Vladimir Karpukhin et al., Dense Passage Retrieval for Open-Domain Question Answering, *EMNLP 2020*.

---

### Meta-Review · Area_Chair_hZ33 · 2023-09-19

**Recommendation:** 4

**Metareview:**

This paper proposes a method for incorporating time-sensitive information into pretrained language models. The authors extract time-stamps, encode them as a graph and propose the Time Relation Classification (TRC) pretraining objective. The paper shows promising results on temporal QA and, interestingly, offers better modeling of complex dependencies. This is an important advancement since temporal reasoning is indeed problematic for SOTA LMs.

One reason the paper got initially moderate scores with the reviewers was due to problematic evaluation: all the three reviewers raise concerns wrt baseline. This, however, seems more like a presentation issue that has been resolved with the authors' thorough rebuttal (cf. discussion). I would strongly suggest, however, that the authors incorporate the relevant discussion into the next version to improve clarity.

Another issue, raised by all the reviewers in some aspect, is the approach for extracting temporal information from text (pipeline). As it seems, the authors only extract explicit timestamps and then merge them rather straightforwardly. They do not assign temporal information to sentences with no explicit timestamps and do not model non-trivial cases of temporal expressions (e.g., "later") -- for example, "X submitted a paper to EMNLP-2023. They resubmitted it again to ACL" would be problematic as it seems? However, even with a simplistic approach to extracting temporal expressions, the model achieves interesting results.

---

### Decision · Program_Chairs · 2023-10-07

**Decision:**

Accept-Main

**Comment:**

This paper proposes a method for incorporating time-sensitive information into pretrained language models. The authors extract time-stamps, encode them as a graph and propose the Time Relation Classification (TRC) pretraining objective. The paper shows promising results on temporal QA and, interestingly, offers better modeling of complex dependencies. This is an important advancement since temporal reasoning is indeed problematic for SOTA LMs.

One reason the paper got initially moderate scores with the reviewers was due to problematic evaluation: all the three reviewers raise concerns wrt baseline. This, however, seems more like a presentation issue that has been resolved with the authors' thorough rebuttal (cf. discussion). I would strongly suggest, however, that the authors incorporate the relevant discussion into the next version to improve clarity.

Another issue, raised by all the reviewers in some aspect, is the approach for extracting temporal information from text (pipeline). As it seems, the authors only extract explicit timestamps and then merge them rather straightforwardly. They do not assign temporal information to sentences with no explicit timestamps and do not model non-trivial cases of temporal expressions (e.g., "later") -- for example, "X submitted a paper to EMNLP-2023. They resubmitted it again to ACL" would be problematic as it seems? However, even with a simplistic approach to extracting temporal expressions, the model achieves interesting results.